# Association between Self-Stigma and Suicide Risk in Individuals with Schizophrenia: Moderating Effects of Self-Esteem and Perceived Support from Friends

**DOI:** 10.3390/ijerph192215071

**Published:** 2022-11-16

**Authors:** Cian-Ruei Jian, Peng-Wei Wang, Huang-Chi Lin, Mei-Feng Huang, Yi-Chun Yeh, Tai-Ling Liu, Cheng-Sheng Chen, Ya-Ping Lin, Shu-Ying Lee, Ching-Hua Chen, Yun-Chi Wang, Yu-Ping Chang, Yi-Lung Chen, Cheng-Fang Yen

**Affiliations:** 1Department of Psychiatry, Kaohsiung Medical University Hospital, Kaohsiung 80756, Taiwan; 2Department of Psychiatry, School of Medicine, Graduate Institute of Medicine, College of Medicine, Kaohsiung Medical University, Kaohsiung 80708, Taiwan; 3Department of Nursing, Kaohsiung Medical University Hospital, Kaohsiung 80756, Taiwan; 4School of Nursing, University at Buffalo, The State University of New York, Buffalo, NY 14260, USA; 5Department of Psychology, Asia University, Taichung 41354, Taiwan; 6Department of Healthcare Administration, Asia University, Taichung 41354, Taiwan; 7College of Professional Studies, National Pingtung University of Science and Technology, Pingtung 91201, Taiwan

**Keywords:** schizophrenia, suicide, self-stigma, self-esteem, social support, mental health

## Abstract

This cross-sectional study assessed the moderating effects of self-esteem and perceived support from friends on the association between self-stigma and suicide risk in individuals with schizophrenia. We included 300 participants (267 with schizophrenia and 33 with schizoaffective disorder). Suicide risk was assessed using items adopted from the suicide module of the Mini-International Neuropsychiatric Interview; self-stigma was assessed using the Self-Stigma Scale–Short; perceived support from friends was assessed using the Friend Adaptation, Partnership, Growth, Affection, and Resolve Index; and self-esteem was assessed using the Rosenberg Self-Esteem Scale. A moderation analysis was performed to examine the moderating effects of self-esteem and perceived support from friends on the association between self-stigma and suicide risk. The results indicated that self-stigma was positively associated with suicide risk after the effects of other factors were controlled for. Both perceived support from friends and self-esteem significantly reduced the magnitude of suicide risk in participants with self-stigma. Our findings highlight the value of interventions geared toward ameliorating self-stigma and enhancing self-esteem in order to reduce suicide risk in individuals with schizophrenia.

## 1. Introduction

### 1.1. Suicide in Individuals with Schizophrenia

Suicide is a major public health problem worldwide. Approximately 800,000 people die by suicide every year [1]. Individuals with mental illnesses have a significant higher risk of suicide than do those without mental illnesses [2]. Additionally, suicide risk is higher among individuals with psychotic disorders than among those with depressive disorders, bipolar disorders, or substance-related disorders [3]. Meta-analyses have revealed the lifetime prevalence of suicidal ideation, suicide plans, and suicide attempts among individuals with schizophrenia to be 34.5%, 44.3%, and 26.8%, respectively [4,5]. Identifying risk factors for suicide and accordingly developing intervention programs are highly crucial for individuals with schizophrenia [6].

### 1.2. Association between Self-Stigma and Suicide

Individuals with schizophrenia experience not only the negative effects of the mental illness but also stigma in their daily lives [7]. Multiple types of stigma toward schizophrenia, such as public stigma, structural stigma, courtesy stigma, provider-based stigma, and self-stigma, can negatively affect the well-being and recovery of individuals with schizophrenia [8,9,10]. Of these types of stigma, self-stigma has the greatest negative effect on individuals with severe mental illnesses [11]. Self-stigma refers to an individual’s acceptance and internalization of negative stereotypes from the public [9,12]. Self-stigma is associated with increased risks of social anxiety, hopelessness, psychiatric symptoms, and low treatment adherence among individuals with mental illnesses [7,13,14]. Both cross-sectional [15,16] and longitudinal [17,18] studies have revealed a significant correlation between self-stigma and depressive symptoms among individuals with schizophrenia. Although studies have examined the association between self-stigma and depression, few have focused on the association between self-stigma and suicide risk among individuals with schizophrenia.

According to the diathesis–stress model [19] and interpersonal–psychological theory [20], individuals with schizophrenia are biologically vulnerable to suicide [21]. As a psychological stressor, self-stigma may further exacerbate suicide risk among individuals with schizophrenia. Cross-sectional studies have confirmed a significant association between self-stigma and suicide risk among individuals with schizophrenia in Taiwan [6], the Czech Republic [22], and Egypt [23]. A 2-year longitudinal study on disability pensioners with mental illness found that more self-stigma predicted suicidal ideation at baseline and longitudinally [24]. Another longitudinal study found that elevated self-stigma was predicted by history of suicide attempt in individuals with serious mental illness or autism spectrum disorder [25]. However, a cross-sectional study found that the association between suicidal ideation and self-stigma was nonsignificant [26]. Medical professionals should continue studying the relationship between self-stigma and suicidal risk in individuals with schizophrenia and help them to ameliorate self-stigma and its effect on suicide risk [27].

### 1.3. Moderating Effects of Perceived Support from Friends and Self-Esteem on the Association between Self-Stigma and Suicide

Studies have yet to examine moderators of the association between self-stigma and suicide risk among individuals with schizophrenia. Identifying the moderators of this association can facilitate the development of subgroup-specific intervention programs for reducing suicide risk among individuals with schizophrenia. According to ecological systems theory [28], both individual (e.g., self-esteem) and individual–environment interaction (e.g., perceived social from friends) factors contribute to suicide risk [29,30]. Self-esteem is a sense of self-acceptance and respect for oneself, and is based on the human need to be valued or to hold a positive self-evaluation [31]. Low self-esteem is prevalent among individuals with schizophrenia [32] and individuals at ultra-high risk for psychosis [33]. Research also found that self-esteem was significantly lower in individuals with a history of previous suicide attempts than in those with no history after controlling for self-stigma and depression [34,35], indicating that self-esteem has an important role for suicide in individuals with schizophrenia.

Social dysfunction is a core feature of schizophrenia [36]. However, most of previous studies have focused on the role of family support but not on support from friends [37]. Lack of social support from friends [38] is prevalent and contributes to an increased risk of suicide among individuals with schizophrenia [26,39], indicating that perceived support from friends is worthy of further attention in intervention programs for suicide in individuals with schizophrenia.

No study has examined the moderating roles of self-esteem and perceived support from friends in the association between self-stigma and suicide risk among individuals with schizophrenia. Nevertheless, studies have revealed that social support and self-esteem reduce the adverse effects of stigma on suicide risk in other discriminated and marginalized populations, such as people living with human immunodeficiency virus infection [40,41,42], indicating that high self-esteem and social support from friends have the protective effects on reducing suicidal risk for the stigmatized individuals. Whether self-esteem and perceived support from friends have similar moderating effects among individuals with schizophrenia warrants further research.

The present study assessed the association between self-stigma and suicide risk among individuals with schizophrenia and determined whether self-esteem and perceived support from friends could moderate the association between self-stigma and suicide risk. We hypothesized that self-stigma is positively associated with suicide risk among individuals with schizophrenia and that self-esteem and perceived support from friends moderate the association between self-stigma and suicide risk.

## 2. Materials and Methods

### 2.1. Participants and Procedure

This study included individuals who visited the psychiatric outpatient clinics and day-care centers of Kaohsiung Medical University Hospital and two community psychiatric rehabilitation institutions in Kaohsiung, Taiwan, between February and May 2022. The inclusion criteria were as follows: (1) being diagnosed as having schizophrenia or schizoaffective disorder by a board psychiatrist in accordance with the fifth edition of the *Diagnostic and Statistical Manual of Mental Disorders* (*DSM-5*) [43] and (2) being aged between 20 and 70 years. The exclusion criteria were as follows: (1) being diagnosed as having an intellectual disability or substance use disorder other than nicotine use disorder by a board psychiatrist and (2) having a history of brain injury or any severe physical disease that would reduce the ability of the individual to understand the study aim, procedure, or interview. A total of 362 individuals were confirmed to be eligible by eight psychiatrists and were invited to participate in this study. Of these individuals, 62 refused and 300 agreed to participate in this study. All participants provided written informed consent before the start of the interview. Two trained research assistants conducted face-to-face interviews with the participants in interview rooms. During the interviews, the participants’ sociodemographic and illness characteristics, suicide risk, self-stigma, perceived support from friends, and self-esteem were assessed using a research questionnaire. Each interview lasted 20 to 30 min, with the exact duration varying for each participant. Psychiatrists evaluated participants’ psychiatric symptoms based on the Positive and Negative Syndrome Scale (PANSS) [44]. All participants were assured that their responses would remain confidential. This study was approved by the Institutional Review Board of Kaohsiung Medical University Hospital, Taiwan (KMUHIRB-SV(I)-20210096).

### 2.2. Measures

The participants’ suicide risk (dependent variable), self-stigma (independent variable), self-esteem and perceived support from friends (moderators), and psychiatric symptoms and sociodemographic factors (covariates) were assessed on the basis of the questionnaire responses.

#### 2.2.1. Suicide Risk

Five items in the suicide module of the Mini-International Neuropsychiatric Interview (MINI) [45] were used to assess the participants’ suicide risk in the preceding month: thinking of death (“Have you ever felt so bad that you wished you were dead?”), wanting to self-harm (“Have you ever wanted to hurt or to injure yourself?”), thinking of suicide (“Have you ever tried to kill yourself?”), having a suicide plan (“Have you ever had a plan to kill yourself?”), and attempting suicide (“Have you ever tried to kill yourself?”). Each item was answered with a “yes” or “no” response. The total number of items that received a “yes” response indicated the severity of suicide risk. Cronbach’s alpha in this study was 0.71.

#### 2.2.2. Self-Stigma

The nine-item Taiwanese version of the Self-Stigma Scale–Short was used to assess the participants’ cognitive, affective, and behavioral self-stigma toward having a mental illness (e.g., “I fear that others would know that I am a mental health consumers”; “I avoid interacting with others because I am a mental health consumers”) [11]. This scale assesses an individual’s level of agreement with the description of each item, with the item rated on a 4-point Likert scale ranging from 1 (*strongly disagree*) to 4 (*strongly agree*). The original version [11] and Taiwanese version of the Self-Stigma Scale–Short [46] were found to have acceptable internal consistency (Cronbach’s alpha = 0.81–0.84 and 0.87, respectively) and satisfactory construct validity (comparative fit index = 0.97 and 0.99, respectively) in a confirmatory factor analysis model. A higher total score indicates a higher level of self-stigma toward having a mental illness. Cronbach’s alpha in this study was 0.88.

#### 2.2.3. Perceived Support from Friends

The five-item Chinese version [47] of the Adaptation, Partnership, Growth, Affection, and Resolve Index (APGAR) [48] was used to assess the participants’ perceived support from friends (e.g., “I am satisfied with the help that I receive from my friend when something is troubling me”). Each item is rated on a 4-point Likert-type scale ranging from 1 (*never*) to 4 (*always*). A higher total score indicates a higher level of perceived support from friends. Cronbach’s alpha in this study was 0.93.

#### 2.2.4. Self-Esteem

The 10-item Rosenberg Self-Esteem Scale (RSES) was used to assess the participants’ global self-esteem (e.g., “On the whole, I am satisfied with myself”) [49]. Each scale item is rated on a 4-point Likert-type scale ranging from 1 (*strongly disagree*) to 4 (*strongly agree*). A higher total score indicates a higher level of global self-esteem. This scale has high reliability and construct validity [49]. Cronbach’s alpha in this study was 0.84.

#### 2.2.5. Psychiatric Symptoms

The 30-item Chinese version of the PANSS was used to assess the severity of the participants’ psychiatric symptoms [44]. The PANSS contains three modules that are used to assess an individual’s positive symptoms (7 items), negative symptoms (7 items), and general psychopathology (16 items). Psychiatrists rate each item on a 7-point Likert-type scale ranging from 1 (*absent*) to 7 (*extreme*). In the present study, the mean scores of the three modules (mean score of positive symptoms subscale + mean score of negative symptoms subscale + mean score of general psychopathology subscale) were summed to represent the severity of the participants’ psychiatric symptoms. A higher total PANSS score indicates more severe psychiatric symptoms. Cronbach’s alpha of the three modules in this study ranged from 0.70 to 0.78.

#### 2.2.6. Sociodemographic and Illness Characteristics

Data regarding the participants’ sex (0 = female; 1 = male), age, education level, monthly disposable income, and years of illness were collected.

### 2.3. Data Analysis

SAS 9.4 (SAS Institute Inc., Cary, NC, USA) was used for all statistical analyses in this study. Sociodemographic data (age, sex, education level, and monthly disposable income), psychiatric symptoms, self-stigma, suicide risk, perceived support from friends, and self-esteem were analyzed using descriptive statistics. Skewness and kurtosis were examined to determine the normality of continuous variables. According to Kim [50], when the sample size is 300 or less, the criteria for normality are that the skewness and Pearson’s kurtosis values must be within ±3.29. Our preliminary analysis revealed that suicide risk was not normally distributed, with the skewness and kurtosis values being 2.90 and 10.12, respectively. We transformed the variable of suicide risk into the log of suicide risk and observed that the skewness and kurtosis values were 1.74 and 2.11, respectively, indicating that normality was a reasonable assumption.

A linear regression analysis model was used to assess the association between self-stigma and suicide risk after the effects of sociodemographic factors, psychiatric symptoms, perceived support from friends, and self-esteem were controlled for. Furthermore, a moderation analysis was performed using the PROCESS v4.0.0 macro, which is based on linear regression modeling [51]. The Johnson–Neyman technique was used to determine whether the conditional effects of self-stigma on the log of suicide risk would differ at specific values of the moderators (self-esteem and perceived support from friends) and to assess the statistical significance of the conditional effects within the range of measurement of the moderators. The significant transition points in the moderation models indicated that the conditional effects of self-stigma on the log of suicide risk differed significantly at specific values of the moderators. A two-tailed *p* value of <0.05 indicated statistical significance.

## 3. Results

Table 1 presents the descriptive statistics of sociodemographic data, psychiatric symptoms, self-stigma, suicide risk, perceived support from friends, and self-esteem as frequencies and proportions or as means and standard deviations (SDs). The participants’ mean age was 45.9 years (SD = 11.7 years). Their mean duration of education was 13.0 years (SD = 2.6 years). Moreover, their mean monthly disposable income was NT$8230.2 (SD = NT$8366.6), mean years of illess was 18.9 years (SD = 10.1 years), and mean PANSS score was 10.6 (SD = 2.2). The mean (SD) scores for self-stigma, suicide risk, perceived support from friends, and self-esteem were 20.0 (5.2), 0.4 (0.9), 13.3 (4.4), and 28.1 (5.5), respectively.

The linear regression analysis model revealed that self-stigma was positively associated with suicide risk after the effects of the other factors were controlled for (Table 2). Our moderation analysis performed to determine the possible moderating effects of self-esteem and perceived support from friends on the association between self-stigma and the log of suicide risk revealed that self-esteem moderated the association. Perceived support from friends and self-esteem significantly reduced the magnitude of suicide risk in participants with self-stigma (coefficients = −0.002, *p* = 0.0496, and −0.003, *p* < 0.001, respectively).

The moderating effects are presented in Figure 1. Having higher perceived support from friends and self-esteem corresponded to a lower intensity of suicide risk among participants with self-stigma. According to the Johnson–Neyman technique, a self-esteem score of ≥28 and perceived friend support ≥11 indicates that the conditional effects of self-stigma and friend support on suicide risk are no longer significant.

## 4. Discussion

This study revealed that self-stigma was significantly associated with suicide risk among participants with schizophrenia. Perceived support from friends and self-esteem moderated the association between self-stigma and suicide by reducing the magnitude of suicide risk among participants with self-stigma.

Similar to the results of previous studies [22,23], our results support the role of self-stigma in increasing suicide risk among individuals with schizophrenia. According to the diathesis–stress model of suicide [2], self-stigma is a psychosocial stressor that can act together with biological vulnerabilities to increase suicide risk among individuals with schizophrenia. Moreover, according to the three-step theory of suicide [52], self-stigma may result in psychological pain and subsequently increase suicide risk. According to the interpersonal–psychological theory of suicide [20], self-stigma can result in social self-isolation and reduce the social support that individuals with schizophrenia need to keep their life going. In addition, self-stigma may lead to aggravated symptoms, such as depression and hopelessness, and further increase suicide risk [35].

According to our review of the literature, our study is the first to identify the moderating effects of self-esteem on the association between self-stigma and suicide risk among individuals with schizophrenia. Individuals with high self-esteem have a sense of self-acceptance and self-respect [31]. Self-esteem develops from individuals’ past experiences and perceptions of social interactions and can serve as a positive personal resource of meaningfulness in life [53]. Self-esteem can also enhance one’s ability to overcome difficulties and stressful life events [54]. Although self-stigma may damage self-value and compromise the level of self-esteem in an individual with schizophrenia [55], the individual may develop self-esteem through multiple experiences in their development history. Self-esteem may reduce the negative effect of self-stigma on psychological well-being and social interaction and thus reduce the magnitude of the association between self-stigma and suicide risk among individuals with schizophrenia.

The present study found that perceived support from friends moderated the association by reducing the magnitude of suicide risk among participants with self-stigma. A previous study revealed that lower help-seeking from friends plays a unique role in increasing suicide risk among individuals with schizophrenia compared with individuals without schizophrenia [39]. Social isolation is also associated with suicide risk among individuals with schizophrenia and first-episode psychosis [56,57]. According to the interpersonal–psychological theory of suicide [58], support from friends can relieve the feeling of being a burden to others, increase the feeling of belonging with others, and reduce hopelessness; therefore, suicide risk can be ameliorated. Because support from friends involves multiple components (e.g., emotional, material, and health-related support), further research is warranted to examine which components related to friend support may contribute to the moderating effects.

### 4.1. Implications

On the basis of our study findings, we propose the following suggestions: First, self-stigma warrants intervention among individuals with schizophrenia. Research has supported that psychoeducation and cognitive behavioral therapy can reduce self-stigma among individuals with schizophrenia [59,60]. However, the reduction of public stigma toward schizophrenia is the fundamental approach toward ameliorating self-stigma among individuals with schizophrenia. Broadening the public’s understanding of schizophrenia and increasing the awareness of prejudices toward individuals with schizophrenia in educational settings, work settings, and the media are crucial steps to help reduce public stigma toward schizophrenia. Second, self-esteem is an intervention target for ameliorating the association between self-stigma and suicide risk in individuals with schizophrenia. Cognitive behavioral therapy can increase the self-esteem of individuals with schizophrenia [60,61]. Third, enhancing social skills and social networks to increase support from friends is necessary to promote the mental health of individuals with schizophrenia.

### 4.2. Limitations

This study has some limitations. First, we used a cross-sectional study design; hence, we could not assess the causality among the measured variables. Second, we collected self-reported data from the participants but did not obtain information from other sources. Third, we recruited participants from psychiatric outpatient units and community psychiatric rehabilitation institutions. Therefore, whether the results of this study can be generalized to individuals who do not visit psychiatric medical units or live in chronic psychiatric wards warrants further examination. Fourth, we used the five items in the suicide module of the MINI [45] to assess the participants’ suicide risk. The original MINI was developed for the purpose of interviewing for the diagnoses of psychiatric disorders but not specifically for assessing suicidal risk. Instruments that specifically assess suicidal risk such as the Columbia Suicide Severity Rating Scale [62] should be considered.

## 5. Conclusions

This study demonstrated a significant association between self-stigma and suicide risk among individuals with schizophrenia and revealed the moderating effects of self-esteem on this association. Self-stigma should be evaluated and managed to reduce suicide risk among individuals with schizophrenia. Moreover, intervention programs for enhancing friend support and self-esteem should be implemented to reduce suicide risk resulting from self-stigma among individuals with schizophrenia.

## Figures and Tables

**Figure 1 ijerph-19-15071-f001:**
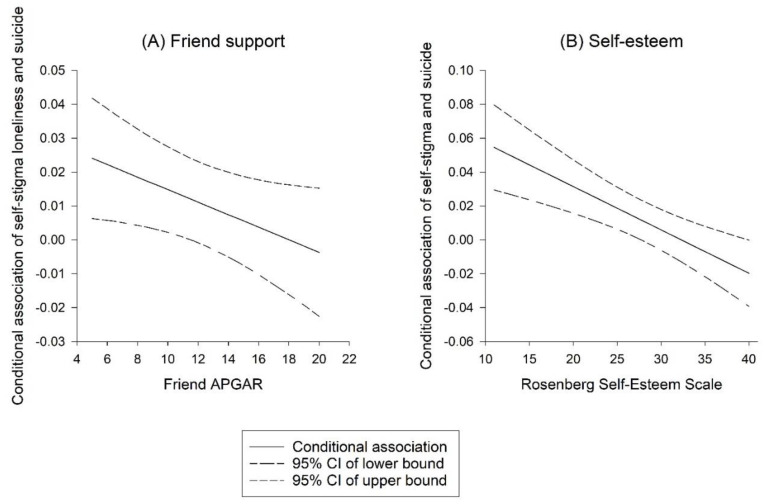
Moderating effects of perceived support from friends and self-esteem on the association between self-stigma and suicide risk among individuals with schizophrenia.

**Table 1 ijerph-19-15071-t001:** Characteristics of participants (*N* = 300).

Variable	*N* = 300
	n (%)
Gender	
Male	139 (46.3%)
Female	161 (53.7%)
	Mean (SD)
Age (year)	45.9 (11.7)
Education (year)	13.0 (2.6)
Money that could be spent freely (NT dollar)	8230.2 (8366.6)
Years of illness (year)	18.9 (10.1)
PANSS score	10.6 (2.2)
Suicide risk	0.4 (0.9)
Self-stigma	20.0 (5.2)
Perceived support from friends	13.3 (4.4)
Self-esteem	28.1 (5.5)

**Table 2 ijerph-19-15071-t002:** Moderating effects of self-esteem and perceived support from friends on the association between self-stigma and suicide risk among individuals with schizophrenia.

	*Model without Moderator*	*Moderation Model*	
		Perceived Support from Friends	Self-Esteem
	*B (se)*	*B (se)*	*B (se)*
Self-stigma	0.011 (0.006) *	0.033 (0.013)	0.083 (0.019) ***
Gender	−0.005 (0.051)	−0.007 (0.050)	<0.001 (0.050)
Age	−0.005 (0.003) *	−0.005 (0.002) *	−0.005 (0.003)
Education	−0.009 (0.010)	−0.009 (0.010)	−0.001 (0.010)
Money that could be spent freely ^a^	<0.001 (0.003) *	<0.001 (0.003)	<0.001 (0.003)
PANSS	0.017 (0.012)	0.017 (0.012)	0.017 (0.012)
Years of illness	0.001 (0.003) *		
Perceived support from friends	−0.004 (0.006) *	0.001 (0.003)	-
Self-esteem	−0.011 (0.006) *	-	0.038 (0.014)
Moderation-term			
Self-stigma by perceived support from friends	-	−0.002 (0.001) *	-
Self-stigma by self-esteem	-	-	−0.003 (0.001) ***

^a^ Data are presented as NT$1000 per unit. * *p* < 0.05, *** *p* < 0.001.

## Data Availability

The data will be available upon reasonable request to the corresponding authors.

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
