# Peer review of "Association between Self-Stigma and Suicide Risk in Individuals with Schizophrenia: Moderating Effects of Self-Esteem and Perceived Support from Friends"

_ijerph, 2022, doi:10.3390/ijerph192215071_

Round 1

Reviewer 1 Report

Thank you for the opportunity to review this article.

Major issue:
- PANSS total score - how is it possible that mean score is 10.6, when the range is 30-210?? This needs to be corrected before publication!

Minor issues:
- you mention looking into positive, negative and general symptoms of the PANSS scale, but there is no indication on the mean scores; do those partial scores correlate with self-esteem/self-stigma? perhaps additional analysis using the five factor model would bring some additional insight with at least a brief view on the depressive symptomatology (of course using CDSS or MADRS would be a better idea)
- relatively low Cronbach's alpha in assessment of suicidality by MINI is not a surprise, perhaps C-SSRS should be considered in future studies
- it would be interesting to see analysis with another variable - 'years of illness', as ongoing schizophrenia leads to deterioration of self-esteeem; the longer the disease, the worse the results would probably be; for now, not including this variable may be considered a limitation

Author Response

We appreciated your valuable comments. As discussed below, we have revised our manuscript with underlines based on your suggestions. Please let us know if we need to provide anything else regarding this revision.

Comment 1
- PANSS total score - how is it possible that mean score is 10.6, when the range is 30-210?? This needs to be corrected before publication!
Response

Thank you four your comment. Because of the different items on the three subscales (7 items in positive symptoms subscale, 7 items in negative symptoms subscales, and 16 items in the general psychopathology subscale), this study transformed the total scores of subscales into mean scores of subscales and summed up them to represent the severity of psychiatric symptoms. Therefore, the score “10.6” was the summed mean score but not the raw score of the PANSS. We revised the description to make the scoring clearer as below. Please refer to line 186-189.

In the present study, the mean scores of the three modules (mean score of positive symptoms subscale + mean score of negative symptoms subscale + mean score of general psychopathology subscale) were summed to represent the severity of the participants’ psychiatric symptoms.

Comment 2
- you mention looking into positive, negative and general symptoms of the PANSS scale, but there is no indication on the mean scores; do those partial scores correlate with self-esteem/self-stigma? perhaps additional analysis using the five factor model would bring some additional insight with at least a brief view on the depressive symptomatology (of course using CDSS or MADRS would be a better idea)
Response

Thank you for your suggestion. We did assess participants’ depressive symptoms by the Center for Epidemiological Studies-Depression Scale (CES-D). However, the association of depressive symptoms with suicidal risk was so strong and made the associations of independent variable (self-stigma), moderators (self-esteem and perceived support from friends), and covariates (demographics and psychiatric symptoms) with suicidal risk nonsignificant. Given that the aim of this study was to examine the association between self-stigma and suicidal risk and the moderators, we did not enter depressive symptoms into linear regression analysis.

Comment 3
- relatively low Cronbach's alpha in assessment of suicidality by MINI is not a surprise, perhaps C-SSRS should be considered in future studies
Response

Thank you for your suggestion. We added it into the revised manuscript as below. Please refer to line 309-313.

Fourth, we used the five items in the suicide module of the MINI [45] to assess the participants’ suicide risk. The original MINI was developed for the purpose of interviewing for the diagnoses of psychiatric disorders but not specifically for assessing suicidal risk. Instruments that specifically assess suicidal risk such as the Columbia Suicide Severity Rating Scale [64] should be considered.”

Comment 4

- it would be interesting to see analysis with another variable - 'years of illness', as ongoing schizophrenia leads to deterioration of self-esteem; the longer the disease, the worse the results would probably be; for now, not including this variable may be considered a limitation

Response

Thank you for your suggestion. We added 'years of illness' into consideration and revised the manuscript based on the results of new linear regression analysis model as below.

Methods

  • “…years of illness were collected” Please refer to line 194.

Results

  • …mean years of illess was 18.9 years (SD = 10.1 years),…” Please refer to line 225 and Table 1.
  • Perceived support from friends and self-esteem significantly reduced the magnitude of suicide risk in participants with self-stigma (coefficients = -0.002, p = 0.0496, and −003, p < .001, respectively).” Please refer to line 233 and 235.
  • “Having higher perceived support from friends and self-esteem corresponded to a lower intensity of suicide risk among participants with self-stigma.” Please refer to line 237.
  • “a self-esteem score of ≥28 and perceived friend support ≥11 indicates that the conditional effects of self-stigma and friend support on suicide risk are no longer significant.” Please refer to line 240.

Discussion

  • Perceived support from friends and self-esteem moderated the association between self-stigma and suicide by reducing the magnitude of suicide risk among participants with self-stigma.” Please refer to line 250.
  • The present study found that perceived support from friends moderated the association by reducing the magnitude of suicide risk among participants with self-stigma.” Please refer to line 275-276.
  • Third, enhancing social skills and social networks to increase support from friends is necessary to promote the mental health of individuals with schizophrenia.” Please refer to line 299-301.

Conclusion

  • “…intervention programs for enhancing friend support and self-esteem should be implemented to reduce suicide risk resulting from self-stigma among individuals with schizophrenia.” Please refer to line 318-320.

Reviewer 2 Report

It is an interesting study. However, the authors should consider the following. 

1. The background should be developed more in detail describing each variable with previous studies.  It is too simple to understand why finding the moderating effect is important. 

2. Measurement items for each variable should be presented in the manuscript. Additionally, each scale for each variable is all different. Why there is so many different scales are used?  It needs further explanation. I don't understand why the dependent variable is measured by yes or no.

3. It is important to address how the data was collected in detail. It is too simple. How the data was recorded is important since it will affect the responses.  

Author Response

We appreciated your valuable comments. As discussed below, we have revised our manuscript with underlines based on your suggestions. We also invited an English-native editor to edit our manuscript. We attached the certificate for the edition. Please let us know if we need to provide anything else regarding this revision.

Comment
1. The background should be developed more in detail describing each variable with previous studies.  It is too simple to understand why finding the moderating effect is important.

Response

Thank you for your suggestion. We added more introductions for the variables examined in this study as below. We also introduced why identifying the moderators is important as below.

“1.2. Association between Self-Stigma and Suicide

…Cross-sectional studies have confirmed a significant association between self-stigma and suicide risk among individuals with schizophrenia in Taiwan [6], the Czech Republic [22], and Egypt [23]. A 2-year longitudinal study on disability pensioners with mental illness found that more self-stigma predicted suicidal ideation at baseline and longitudinally [24]. Another longitudinal study found that elevated self-stigma was predicted by history of suicide attempt in individuals with serious mental illness or autism spectrum disorder [25]. However, a cross-sectional study found that the association between suicidal ideation and self-stigma was nonsignificant [26].” Please refer to line 70-77.

1.3. Moderating Effects of Perceived Support from Friends and Self-Esteem on the Association between Self-Stigma and Suicide

  • “Identifying the moderators of this association can facilitate the development of subgroup-specific intervention programs for reducing suicide risk among individuals with schizophrenia. Please refer to line 84-86.
  • Self-esteem is a sense of self-acceptance and respect for oneself, and is based on the human need to be valued or to hold a positive self-evaluation [31]. Low self-esteem is prevalent among individuals with schizophrenia [32] and individuals at ultra-high risk for psychosis [33]. Research also found that self-esteem was significantly lower in individuals with a history of previous suicide attempts than in those with no history after controlling for self-stigma and depression [34,35], indicating that self-esteem has an important role for suicide in individuals with schizophrenia.” Please refer to line 88-95.
  • “Social dysfunction is a core feature of schizophrenia [36]. However, most of previous studies have focused on the role of family support but not on support from friends [37]. Lack of social support from friends [38] is prevalent and contributes to an increased risk of suicide among individuals with schizophrenia [26,39], indicating that perceived support from friends is worthy of further attention in intervention programs for suicide in individuals with schizophrenia.” Please refer to line 96-101.
  • No study has examined the moderating roles of self-esteem and perceived support from friends in the association between self-stigma and suicide risk among individuals with schizophrenia. Nevertheless, studies have revealed that social support and self-esteem reduce the adverse effects of stigma on suicide risk in other discriminated and marginalized populations, such as people living with human immunodeficiency virus infection [40-42], indicating that high self-esteem and social support from friends have the protective effects on reducing suicidal risk for the stigmatized individuals. Whether self-esteem and perceived support from friends have similar moderating effects among individuals with schizophrenia warrants further research.Please refer to line 102-110.

Comment

  1. Measurement items for each variable should be presented in the manuscript. Additionally, each scale for each variable is all different. Why there is so many different scales are used?  It needs further explanation. I don't understand why the dependent variable is measured by yes or no.

Response

Thank you for your suggestion. We added the explanations for the scales used in this study as below. We also added the example items for each scale.

“The participants’ suicide risk (dependent variable), self-stigma (independent variable), self-esteem and perceived support from friends (moderators), and psychiatric symptoms and sociodemographic factors (covariates) were assessed on the basis of the questionnaire responses.” Please refer to line 142-145.

2.2.1. Suicidal Risk

“…thinking of death (“Have you ever felt so bad that you wished you were dead?”), wanting to self-harm (“Have you ever wanted to hurt or to injure yourself?”), thinking of suicide (“Have you ever tried to kill yourself?”), having a suicide plan (“Have you ever had a plan to kill yourself?”), and attempting suicide (“Have you ever tried to kill yourself?”).” Please refer to line 149-153.

2.2.2. Self-stigma

“…(e.g., “I fear that others would know that I am a mental health consumers”; “I avoid interacting with others because I am a mental health consumers”)” Please refer to line 159-160.

2.2.3. Perceived Support from Friends

“...(e.g., “I am satisfied with the help that I receive from my friend when something is troubling me”).” Please refer to line 171-172.

2.2.4. Self-esteem

“...(e.g., “On the whole, I am satisfied with myself”)” Please refer to line 177.

2.2.5. Psychiatric Symptoms

In the present study, the mean scores of the three modules (mean score of positive symptoms subscale + mean score of negative symptoms subscale + mean score of general psychopathology subscale) were summed to represent the severity of the participants’ psychiatric symptoms.” Please refer to line 186-189.

Comment

  1. It is important to address how the data was collected in detail. It is too simple. How the data was recorded is important since it will affect the responses.  

 Response

Thank you for your suggestion. We added the explanations for the process of collecting the data in this study as below. Please refer to line 132-139.

Two trained research assistants conducted face-to-face interviews with the participants in interview rooms. During the interviews, the participants’ sociodemographic and illness characteristics, suicide risk, self-stigma, perceived support from friends, and self-esteem were assessed using a research questionnaire. Each interview lasted 20 to 30 min, with the exact duration varying for each participant. Psychiatrists evaluated participants’ psychiatric symptoms based on the Positive and Negative Syndrome Scale (PANSS) [44]. All participants were assured that their responses would remain confidential.

Reviewer 3 Report

The paper is superb, and is of great importance. The connection between suicide-risk and self-stigma is known intuitively and is well studied (for instance "Self-stigma and suicidality: a longitudinal study, Self-stigma and suicidality: a longitudinal study", Nathalie Oexle and others), as the authors mention themselves (lines 73-74), but any further research that can add and prove it is welcome. And as the paper emphasizes, "Identifying the moderators of this association can facilitate the development of subgroup-specific intervention programs for reducing suicide risk among individuals with schizophrenia". I am waiting to see further study of results of the awareness of likewise studies.

Author Response

We appreciated your valuable comments. As discussed below, we have revised our manuscript with underlines based on your suggestions. Please let us know if we need to provide anything else regarding this revision.

Comment
The paper is superb, and is of great importance. The connection between suicide-risk and self-stigma is known intuitively and is well studied (for instance "Self-stigma and suicidality: a longitudinal study, Self-stigma and suicidality: a longitudinal study", Nathalie Oexle and others), as the authors mention themselves (lines 73-74), but any further research that can add and prove it is welcome.

Response

Thank you for your suggestion. We added more results of previous studies on the association between self-stigma and suicide in individuals with severe mental illness as below. Please refer to line 70-77.

Cross-sectional studies have confirmed a significant association between self-stigma and suicide risk among individuals with schizophrenia in Taiwan [6], the Czech Republic [22], and Egypt [23]. A 2-year longitudinal study on disability pensioners with mental illness found that more self-stigma predicted suicidal ideation at baseline and longitudinally [24]. Another longitudinal study found that elevated self-stigma was predicted by history of suicide attempt in individuals with serious mental illness or autism spectrum disorder [25]. However, a cross-sectional study found that the association between suicidal ideation and self-stigma was nonsignificant [26].

New citations

  1. Dubreucq, J.; Plasse, J.; Franck, N. Self-stigma in serious mental illness: A systematic review of frequency, correlates, and consequences. Schizophr. Bull. 2021, 47, 1261-1287. doi: 10.1093/schbul/sbaa181.
  2. Oexle, N.; Rüsch, N.; Viering, S.; Wyss, C.; Seifritz, E.; Xu, Z.; Kawohl, W. Self-stigma and suicidality: a longitudinal study. Eur. Arch. Psychiatry Clin. Neurosci. 2017, 267, 359-361. doi: 10.1007/s00406-016-0698-1.
  3. Dubreucq, J.; Plasse, J.; Gabayet, F.; Faraldo, M.; Blanc, O., Chereau, I.; Cervello, S.; Couhet, G.; Demily, C.; Guillard-Bouhet, N.; et al. Self-stigma in serious mental illness and autism spectrum disorder: Results from the REHABase national psychiatric rehabilitation cohort. Eur. Psychiatry 2020, 63, e13. doi: 10.1192/j.eurpsy.2019.12. 
  4. Collett, N.; Pugh, K.; Waite, F.; Freeman, D. Negative cognitions about the self in patients with persecutory delusions: An empirical study of self-compassion, self-stigma, schematic beliefs, self-esteem, fear of madness, and suicidal ideation. Psychiatry Res. 2016, 239, 79-84. doi: 10.1016/j.psychres.2016.02.043.

Comment

As the paper emphasizes, "Identifying the moderators of this association can facilitate the development of subgroup-specific intervention programs for reducing suicide risk among individuals with schizophrenia". I am waiting to see further study of results of the awareness of likewise studies.

Response

Thank you for your support. We will continue studying this important issue.

Round 2

Reviewer 1 Report

Dear Authors,

Thank you for the explanations and corrections.

I see no other issues.

Kind regards and best of luck.